# Hog1 Controls Lipids Homeostasis Upon Osmotic Stress in *Candida albicans*

**DOI:** 10.3390/jof6040355

**Published:** 2020-12-10

**Authors:** Carmen Herrero-de-Dios, Elvira Román, Jesús Pla, Rebeca Alonso-Monge

**Affiliations:** 1Servicio de Bioquímica, Hospital Universitario Ramón y Cajal, Ctra. Colmenar Km 9, 28034 Madrid, Spain; cmerrero@gmail.com; 2Departamento de Microbiología y Parasitología-IRYCIS, Facultad de Farmacia, Universidad Complutense de Madrid, Plaza de Ramón y Cajal s/n, 28040 Madrid, Spain; elvirarg@ucm.es (E.R.); jpla@ucm.es (J.P.)

**Keywords:** HOG (High Osmolarity Glycerol) pathway, osmotic stress, lipids homeostasis, ergosterol, MAP kinase, *Candida albicans*

## Abstract

As opportunistic pathogen, *Candida albicans* adapts to different environmental conditions and its corresponding stress. The Hog1 MAPK (Mitogen Activated Protein Kinase) was identified as the main MAPK involved in the response to osmotic stress. It was later shown that this MAPK is also involved in the response to a variety of stresses and therefore, its role in virulence, survival to phagocytes and establishment as commensal in the mouse gastrointestinal tract was reported. In this work, the role of Hog1 in osmotic stress is further analyzed, showing that this MAPK is involved in lipid homeostasis. The *hog1* mutant accumulates lipid droplets when exposed to osmotic stress, leading to an increase in cell permeability and delaying the endocytic trafficking routes. Cek1, a MAPK also implicated in the response to osmotic challenge, did not play a role in lipid homeostasis indicating that Hog1 is the main MAP kinase in this response. The alteration on lipid metabolism observed in *hog1* mutants is proposed to contribute to the sensitivity to osmotic stress.

## 1. Introduction

*Candida albicans* is a human commensal able to cause a great variety of infections ranging from superficial to systemic [1]. Remarkably, *C. albicans* is the major cause of fungemia in hospitals of developed countries [2,3], although other *Candida* species are increasing currently the incidence [4,5]. Both as a commensal and during invasion, *C. albicans* need to face different external conditions in order to survive. Signal transduction pathways mediated by Mitogen Activated Protein Kinases (MAPK) are one of the main mechanisms by which eukaryotic cells respond to extracellular stimuli to adapt and survive [6,7,8]. These MAPK pathways consist on a module of three kinases (named MAP kinase kinase kinase or MAPKKK; MAP kinase kinase or MAPKK; and a MAP kinase or MAPK) that became activated by sequential phosphorylation in response to different signals through other kinases, two-component systems or heterotrimeric G proteins. As a consequence, the activated MAPK triggers an adaptive response regulating the transcription of target genes or modulating enzymatic activities. These signaling mechanisms are conserved through the evolution from simplest (unicellular) to more complex organisms. Each MAPK pathway is activated by a certain set of stimuli and induces the appropriate response.

So far, four MAPKs have been identified in *C. albicans*. Mkc1 was the first MAPK identified and characterized in *C. albicans* [9]. This MAPK belongs to the cell wall integrity (CWI) pathway and is involved in cell wall biogenesis. Later, it was reported to play a role in biofilm formation [10] and in the response to different stresses among them oxidative or osmotic stress [11]. In vivo, the *mkc1* mutant exhibits reduced virulence in a systemic infection model in mice, reinforcing the relevance of this MAPK during infection [12].

Cek1 was identified in a screening looking for genes interfering with pheromone-mediated cell cycle arrest [13]. Later on, this MAPK was implicated in invasive hyphal growth, mating efficiency, cell wall construction, quorum sensing, β-glucan exposure and virulence in a systemic murine model as well as in the insect model *Galleria mellonella* [14,15,16,17,18]. Sharing upstream elements, Cek2 is the least studied MAPK in *C. albicans*. It belongs to the mating pathway although a double *cek1 cek2* deletion is required to complete impaired mating in this yeast. Overexpression of *CEK2* indicates that this MAPK plays redundant roles with Cek1 in cell wall biogenesis and resumption of growth from stationary phase [19]; Cek1 (contrary to Cek2) is marginally involved in the osmotic stress response [20,21]. Although these three MAPKs become activated by different stress conditions, Hog1 is the canonical Stress Activated MAPK or SAPK in *C. albicans*. Hog1 is involved in adaptation to osmotic stress but is also involved in the response to oxidative stress, metals, metalloids, and nitrosative stress among others [22,23,24,25]. Hog1 also participates, in the yeast-to-hypha transition, cell wall biogenesis, oxidative metabolism, virulence, and commensalism [26,27,28] and therefore, plays an essential role in *C. albicans* physiopathology. The role of Hog1 in oxidative stress has attracted attention since the immune system uses oxidative burst to destroy pathogens. Interestingly, *hog1* mutants are able to partially respond to oxidative challenges despite its increased sensitivity [29].

Hyperosmotic stress has been extensively studied in *S. cerevisiae* since this stress is important to survive on the leaves of the vines and in the must where the sugar and solute concentration is very high or during fermentation. In an opportunistic pathogen such as *C. albicans,* the study of the response to osmotic stress has not been so deeply analyzed. Nevertheless, the capability to properly respond to an increase external osmolarity can be crucial to survive dehydration through large intestine transit as commensal or in the kidneys during glomerular filtration as pathogen. In this work, we have explored the survival of *cek1* and *hog1* mutants under hyperosmotic conditions, leading to the role played by Hog1 in lipid homeostasis and endocytic trafficking.

## 2. Materials and Methods

### 2.1. Strains and Growth Conditions

Yeast strains were routinely grown in YPD medium (1% yeast extract (Conda/Pronadisa, Madrid, Spain), 2% bacteriological peptone (Conda/Pronadisa, Madrid, Spain), 2% glucose (D(+)-glucose anhidra, PanReac AppliChem, Barcelona, Spain)) at 37 °C. Overnight cultures were refreshed to pre-warmed medium to an optical density of 0.1 (A_600 nm_) and experiments were performed when cultures reached an optical density of 1 (A_600 nm_) when exponential phase cells were required. An O.D. of 1 corresponds approximately to 2.5 × 10^7^ cell/mL. *C. albicans* strains used in the manuscript are listed in Table 1.

### 2.2. Viability Assays

Viability assays by counting CFUs were performed by exposing exponential growing cell in YPD liquid medium to osmotic stress. For this purpose, NaCl (PanReac AppliChem, Barcelona, Spain) was added to the cultures to 1 M final concentration and flasks were shaken until powder was completely dissolved. Then, 1 mL samples were collected at different times which were washed with PBS (phosphate-buffered saline BD FACS Flow, from Becton-Dickinson Biosciences, Madrid, Spain) to seed different decimal dilutions in solid YPD medium. Due to the different viability of the mutants, a test experiment was previously carried out to determine the appropriate dilution to obtain 100 to 300 colonies per plate in each strain [31]. Parallel cultures were also maintained without sodium chloride to verify that the loss of viability was due to stress caused by NaCl.

### 2.3. Staining with Propidium Iodide

To estimate cell viability and study the permeability of the plasma membrane, samples were collected at different incubation times with NaCl, washed twice with PBS and the cells were stained with propidium iodide (IP) at 0.005% final concentration [32,33]. After staining, the percentage of stained cells was measured by flow cytometry (Guava easyCyte 5 Benchtop Flow Cytometer, Millipore Merck Darmstadt, Germany). The IP has an excitation λ of 488 nm and an emission λ of 630 nm, so the blue laser was used, which has an excitation λ of 488 nm. Data obtained with the Incyte program of the cytometer were plotted using the GraphPad Prism version 7.00 for Windows, GraphPad Software, San Diego, California USA, www.graphpad.com.

### 2.4. Study of Lipid Content

Nile Red is a fluorochrome that emits fluorescence at different wavelengths depending on the polarity of the lipids. Fluorescence signal was quantified in yellow (FL-2), green (FL-1), and red (FL-3) channels. The flow cytometer blue laser (From Guava easyCyte 5 Benchtop Flow cytometer Millipore) was used with an excitation λ of 488 nm. For staining with this fluorochrome, cells were washed with PBS a couple of times and Nile Red (Sigma-Aldrich Inc. St. Louis, MO, USA) (1 mg/mL in methanol) was added in 1 mL of culture resuspended in PBS for quantification by flow cytometry and for image capture with the confocal microscope MRC Bio-Rad (Bio-Rad Laboratories, Hercules, CA, United States).

For the ergosterol staining, we used the filipin stain (Sigma-Aldrich Inc. St. Louis, MO, USA) (excitation λ 380 nm, emission λ 510 nm) [34]. The cells with the fluorochrome (10 μg/mL Filipin to 10^7^ cells) were incubated for 10 min at 37 °C in darkness, and then pictures were taken directly under the fluorescence microscope.

### 2.5. Study of Endocytosis

The marker FM4-64 (Molecular Probes, Thermo Fisher Scientific, Eugene, OR, USA) allowed us to stain the plasma membrane and internal membranes. The cells were harvested during the experiment and the dye was added directly to a final concentration of 40 μM and incubated in general for one hour, otherwise indicated, at 37 °C for the complete internalization of the compound allowing the visualization of the endocytic pathway. Subsequently, the samples were visualized under a fluorescence microscope, considering the excitation and emission wavelengths of 558 and 734 nm, respectively.

### 2.6. Fluorescence Microscopy

Fluorescence microscopy was carried out on a Nikon Eclipse TE2000-U Phase Contrast DIC Fluorescence Inverted microscope (Nikon, Tokyo, Japan) at 100× magnification. Images were captured by a Hamamatsu ORCA-ER CCD camera using AquaCosmos 1.3 software (Hamatmatsu Photonic KK, Hamatmatsu City, Japan). All images were processed identically and mounted using Adobe Photoshop 7.0.

## 3. Results

### 3.1. Hog1 Controls Cell Permeability upon Osmotic Stress

Previous work in *S. cerevisiae* reported that HOG pathway was not crucial for cell viability in the first hours after exposure to hyperosmotic challenge, but it was for efficient recovery and subsequent proliferation under osmostress [35]. While the role of Hog1 in *C. albicans* has been reported [29,36] in the response to osmotic stress, the effect on viability of hyperosmotic stress has not been further characterized. We therefore analyzed the viability of *hog1* mutants upon osmotic challenge. For this purpose, wild type and *hog1* mutant cells were incubated in liquid YPD supplemented with 1 M NaCl and samples were taken at different time points. Samples were spread on YPD plates for CFUs counting and, in parallel, stained with IP to determinate cell viability by flow cytometry. CFUs hardly increased for the *hog1* mutant (Figure 1a) while wild type strain was able to exponentially increase the CFUs relatively soon. The *cek1* and *cek1 hog1* mutants were also included in the study since Cek1 has been previously involved in the osmotic stress response [20,21]. The *cek1* single mutant was able to grow similarly to the wild type strain although with a slight delay. The *cek1 hog1* mutant behaved similar to the *hog1* single mutant scarcely increasing the CFUs: this behavior may indicate a loss of viability or a delay in cell growth resumption. The fact that the CFUs number did not increase along time in the *hog1* and *cek1 hog1* mutants could be due to alteration in cytokinesis as previously reported [20,26]. When, the percentage of IP+ cells was quantified, the *hog1* single mutant exhibited a 60% IP+ cells in the first hour of incubation in the presence of 1 M NaCl reaching the 91.8% IP+ at 8 h (Figure 1b). Similarly, *cek1 hog1* double mutant exhibited higher cell permeability reaching 83.7% IP+ cells after 6 h upon hyperosmotic conditions. The *cek1* mutant behaved as the wild type strain (Figure 1b). The fact that an elevated percentage of cells lacking the *HOG1* gene were stained with IP at short time upon salt challenge may correlate to an increase of cell permeability. These results indicate that both Cek1 and Hog1 cooperate to ensure cell proliferation upon osmotic challenge while Hog1 also mediates the permeability to IP.

### 3.2. Hog1 Mediates Lipids Homeostasis Upon Osmotic Stress

Cell permeability can increase due to the cell death (and subsequent loss of viability) or due to an alteration of lipids in the membrane of the cells [37,38]. Since osmotic stress alters the growth of *hog1* mutants rather that cell viability we quantified the lipid content using the lipophilic dye Nile Red. This dye is used to localize and quantify lipids in different biological systems [39]. When the fluorescence of Nile Red-stained cells is examined at wavelengths of 580 nm or less (FL-2-yellow channel) the signal is due to the interaction with neutral lipids (hydrophobic environment). On the other hand, when the fluorescence is detected at emission wavelengths greater than 590 nm (FL-3-Orange/Red-channel) the fluorescence corresponds to the interaction of Nile Red with phospholipid-rich environments [40]. Figure 2a shows the mean fluorescence intensity emitted in the presence of 1 M NaCl along time in wild type and *hog1* mutant strain. Deletion of the *HOG1* gene caused an increase in the fluorescence signal in both channels (FL-2 and FL-3) indicating an increase in the amount of both polar and neutral lipids. The higher increase was detected at FL-3 suggesting an increase in the intracellular membranes. This augment of intracellular membranes was visualized when Nile Red-stained cells were observed under confocal microscopy (Figure 2b). The control strain displayed red-emitting structures corresponding to phospholipid-rich intracellular membrane. Nevertheless, in the *hog1* mutant these intracellular structures emitted fluorescence detected in green indicating that these phospholipid-rich intracellular membrane can be charged with neutral lipids such as triacylglycerols or cholesteryl esters in a higher amount compared to wild type cells [41]. The fluorescence signal of these lipophilic structures was significantly higher in the *hog1* mutant compared to wild type grown in the same conditions. No differences were observed when cells were incubated in the absence of NaCl (data no shown). The accumulation of lipophilic structures upon osmotic shock depends on an intact HOG pathway suggesting a role of this MAPK in lipids biosynthesis.

Filipin was used to detect the ergosterol under fluorescence microscopy. Filipin is a naturally fluorescent polyene antibiotic that binds sterols and therefore, stains the ergosterol of the fungal membrane. In the wild type strain growth in standard conditions (YPD at 37 °C), the fluorescence was observed mainly associated with internal membranes, as previously described, due to the rapid re-cycling of the membrane lipids. The fluorescence detected in the same strain incubated in the presence of 1 M NaCl remained in the plasma membrane non-homogenously distributed (Figure 2c) [42]. Osmotic stress induced an increase in the fluorescent signal in mutant lacking Hog1 or mutant defective in Cek1/Hog1-mediated pathways. This signal was detected mainly intracellularly suggesting that ergosterol became accumulated in vacuoles or cytoplasmic structures either as precursors or due to a rapid turn-over. These results indicate that HOG pathway may control ergosterol homeostasis and lipids biosynthesis.

### 3.3. Osmotic Stress Delays the Endocytic Trafficking in *hog1* Defective Mutants

Filipin-ergosterol complexes has been reported to co-localize with plasma membrane compartments (MCC) involved in endocytosis [43]. Moreover, we detected these complexes in intracellular compartments (Figure 2c); we therefore analyzed intracellular trafficking in the mutants under study by staining with the lipophilic compound FM4-64. FM4-64 is a vital dye that allows the visualization of the dynamism of the vacuolar membrane and endocytosis in yeast [44]. This agent stains specifically the compartments of the endocytic pathway as well as the vacuolar membrane. FM4-64 compound was added to control cultures, that is without osmolyte addition. In the wild type strain, FM4-64 localized at the vacuolar membrane 1 h after its addition while the *hog1* mutant required 2 h to reach the vacuolar compartments. Similarly, both *cek1* single and *cek1 hog1* double mutants also exhibited a delay in endocytic trafficking (Figure 3a). A lower fluorescence signal was detected in the plasma membrane but still distinguishable after 1 h in both cases, *cek1* and *cek1 hog1* mutants.

Later, exponentially growing cells were challenged with NaCl to 1 M and incubated at 37 °C during 20 h, then, FM4-64 compound was added to the medium to follow the endocytic route. After 1 h of incubation, most of the FM4-64 dye was localized into the vacuole in the wild type strain. In the *hog1* mutant an increased fluorescence was observed according to the increased lipid content and delay 4 h to localize at vacuolar membrane, moreover an elevated number of small intracellular vesicles could be observed, probably due to vacuolar system alterations (Figure 3b). In the case of the *cek1* mutant, the dye reached the vacuolar membrane after 2 h of incubation showing a similar vacuolar fragmentation. The *cek1 hog1* double mutant displayed altered cell morphology and increased fluorescence analogous to the *hog1* mutant. In conclusion, Hog1 and Cek1 control endocytic trafficking during osmotic stress adaptation and under normal growth conditions.

## 4. Discussion

The adaptation to environmental changes allows cells to survive in nature. The response to osmotic stress is complex and crucial for unicellular microorganisms, since the proper adaptation to external osmotic changes will allow cells to survive in the environment either within or outside the host. For example, the ability of *Listeria monocytogenes* to persist in food processing environments correlates to its osmotolerance [45]. In eukaryotes, MAPK signaling transduction pathways mediate this adaptation, leading to specific and adequate responses. The role of the MAP kinase Hog1 in the response to osmotic shock is essential and has been extensively study in *S. cerevisiae* and other organisms [46,47,48,49]. In *C. albicans* two MAP kinases, Hog1 and Cek1 have been involved in the response to hyperosmotic conditions [20,21] being the double mutant *cek1 hog1* highly hypersensitive to osmostress. Although, *cek1* mutant alone did not show osmosensitivity on solid media, a slight delay of resumption of growth was detected when the assay was performed in liquid medium via CFUs count (Figure 1a). IP stain revealed that *hog1* mutant but not *cek1* mutant increased significantly (and rapidly) cell permeability. Although both assays, CFUs count and IP stain suggest that the *hog1* mutant lost viability shortly upon salt addition, longer incubation period on hyperosmotic solid media indicate that *hog1* mutant cells survive this condition (data not shown) similarly to the results reported in *S. cerevisiae* [35]. Then, IP stain implies an increase of permeability that can be independent of cell death but due to an increase of lipid content. Polar membrane phospholipids augment considerably in *hog1* mutant in the presence of 1 M NaCl in parallel with an increase of neutral lipids. Other microorganisms such as microalgae also accumulate lipids in the presence of salts stress which has potential use in the production of biofuels. In the case of the microalgae, *Chlamydomonas reinhardtii*, this increase of neutral lipids was reported to depend on MAPK activation [50,51].

In *S. cerevisiae*, Manzanares-Estrede and co-workers [52] reported that Hog1 controls the expression of fatty acid mobilization and peroxisomal beta-oxidation upon salts addition or glucose deprivation. Enzymes involved in fatty acid mobilization are localized in internal lipid storage and peroxisomes. In response to salt challenge, *S. cerevisiae* cells switch from a fermentative metabolism to fatty acid oxidation and mitochondrial respiration to cover the energetic needs during the environmental change [52]. This process depends, in part, on the Hog1 activation. Here, we reported an increase on internal lipid particles in the *C. albicans hog1* mutant in response to a hyperosmotic shock. This effect can be attributable to the failed attempt of mobilize these lipids due to the lack of Hog1 activation. Moreover, the role of Hog1 in the oxidative metabolism was previously reported in *C. albicans* [53]. Hog1 may play a central role in adapting metabolism to external conditions. In response to salts challenge, as well as in the presence of non-fermentable carbon sources, *C. albicans* cells may require fatty acid oxidation to cope with the energetic requirements.

Membrane lipids are also important in signaling and for the maintenance of signal fidelity. Phospholipids play an important role in maintaining MAP kinase signaling specificity, this phospholipids can be localized either in the plasma membrane [54] or endomembranes [55]. The signaling role is played mainly by phosphoinositides (PIP), which are rapidly modified by kinases and phosphatases [56]. These PIP mark organelles and provide specific binding surfaces for peripheral membrane proteins. In *S. cerevisiae*, the phosphatidylinositol 4-kinase Pik1 locates fundamentally in the Golgi and allows the activation of Hog1 in response to osmotic stress through Opy2 [55]. In *C. albicans*, Hog1 could have a repressive role in the synthesis of membrane lipids and/or PIP kinases/phosphatases activity. Perhaps the increase in membrane lipids (polar lipids) in *hog1* mutants in response to osmotic stress is a compensatory mechanism to try to recruit signaling elements and activate Hog1. In the case of the mutant *hog1*, there would be no repression of synthesis and therefore an increase in membrane lipids occurs.

Within the endolysosomal system, PIP lipids determine the fate of endosomes and vacuoles. In *S. cerevisiae*, the phosphatidylinositol 3,5-bisphosphate, PI-3,5-P_2_, is critical for lysosomal membrane homeostasis during acute osmotic stress and for lysosomal signaling [57]. In the present work, we report that, upon salts challenge, a delay in the arrival of FM4-64 dye to the vacuolar membrane occurs in the wild type strain. This fact suggests that endocytic trafficking also responds to environmental conditions in *C. albicans*. Hog1 and Cek1 MAPKs may control this trafficking either controlling the expression or activity of PIP modifying enzymes, controlling lipids composition or posttranslational modifications of proteins. Any of these functions may cause the phenotypes reported to impair the growth upon hyperosmotic stress.

In summary, the osmostress induces drastic changes in yeast cells including metabolic adaptations, signaling lipid modifications and delay in endocytic trafficking. Both Hog1 and Cek1 MAPKs control these processes in *C. albicans*.

## Figures and Tables

**Figure 1 jof-06-00355-f001:**
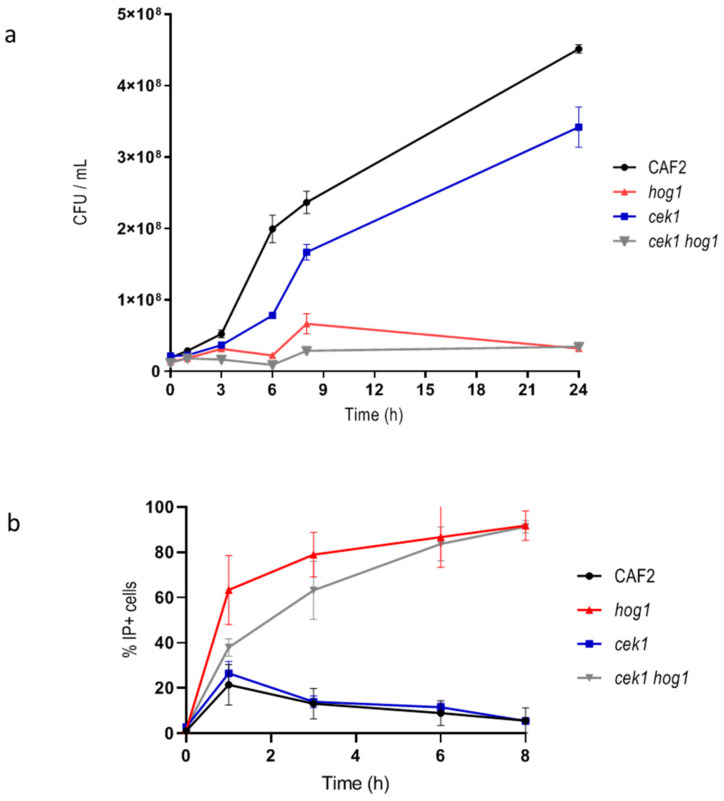
Quantification of cell viability upon hyperosmotic challenge. (**a**) Graph showing the cellular growth by CFUs / mL count along the time of the indicated strains. (**b**) Graph represents IP+ cells quantified by flow cytometry along the time. Graphs show the mean and the standard error of the mean of at least three independent experiments.

**Figure 2 jof-06-00355-f002:**
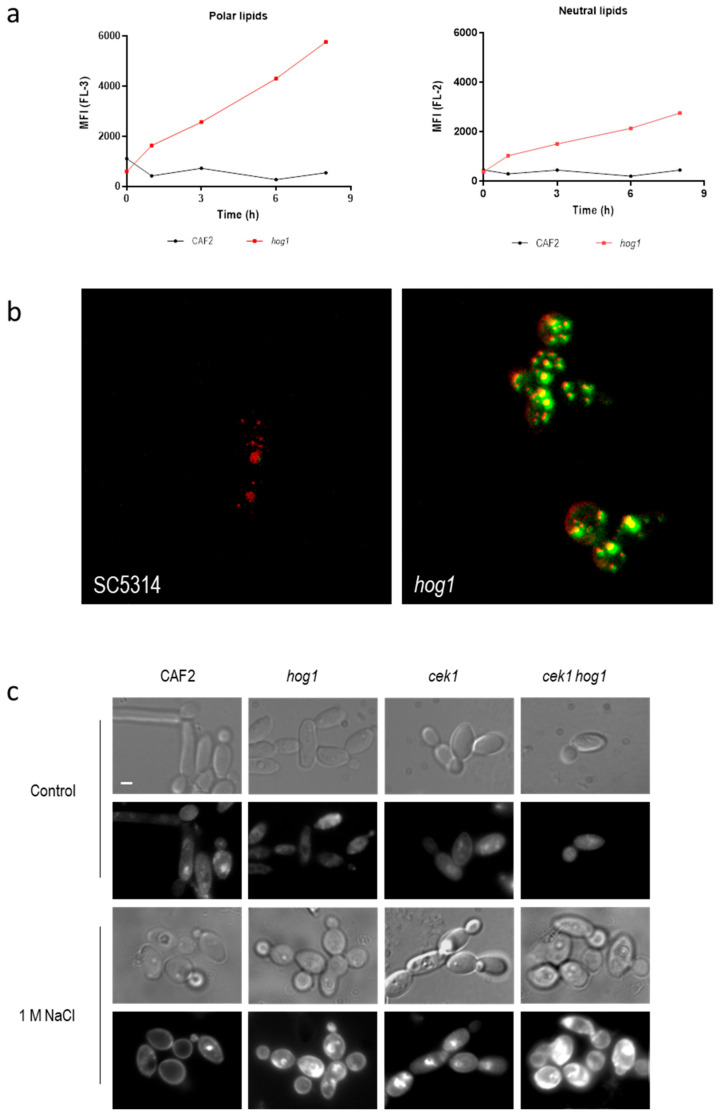
Detection of intracellular lipids upon hyperosmotic challenge. (**a**) Exponentially growing cells were exposed to 1 M NaCl and stained with Nile Red. Fluorescence in FL-2 and FL-3 channels was quantified and mean fluorescent intensity (MFI) belonging to a standard assay is represented along the time. Fluorescence in FL-3 correlates with polar lipid quantification (left panel) while fluorescence of FL-2 correlates with neutral lipids (right panel). (**b**) Visualization of wild type and *hog1* mutant cells exposed to 1 M NaCl and stained with Nile Red with a confocal microscope. Red and green fluorescence is shown (co-localization is seen in yellow). (**c**) Detection of ergosterol by filipin stain. *C. albicans* cells growing in YPD (control) or YPD supplemented with 1 M NaCl for 24 h were stained with filipin (10 μg/mL for 10^7^ cells) and observed under phase contrast microscopy (odd rows) or fluorescence microscopy (pair rows). Bars, 1 μm.

**Figure 3 jof-06-00355-f003:**
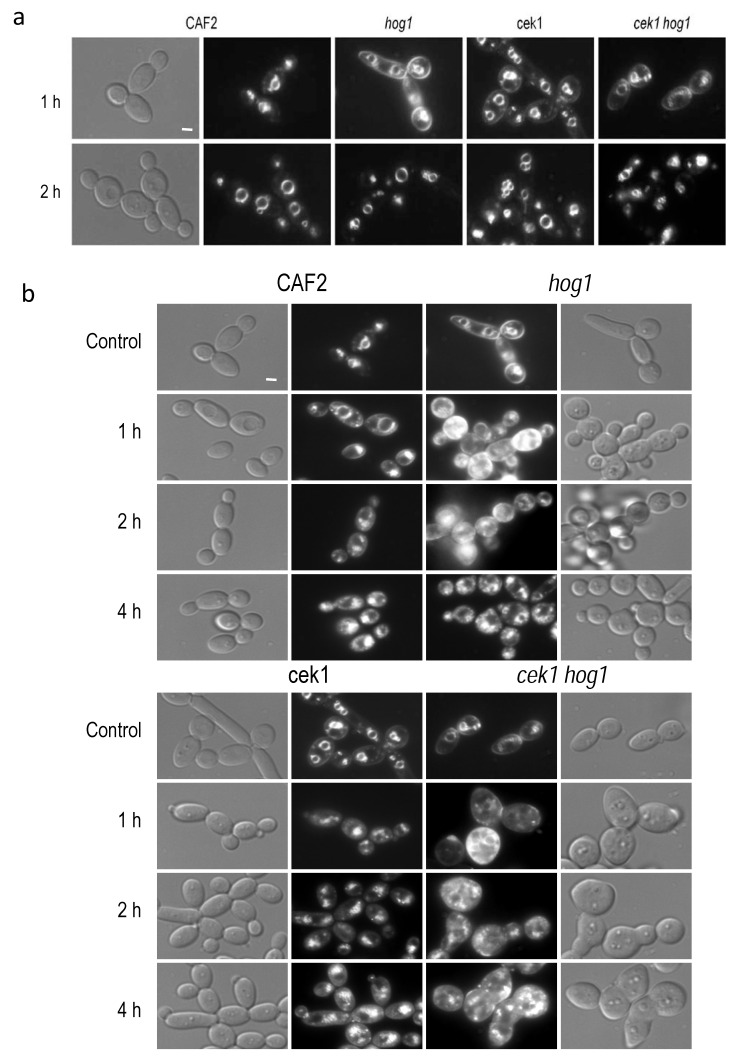
Endocytic trafficking. (**a**) FM4-64 dye was added to cells growing in YPD at 37 °C to visualize its internalization. Pictures were taken 1 and 2 h after dye addition using a Normasky and fluorescence microscopy. (**b**) Cells incubated for 20 h in YPD plus 1 M NaCl at 37 °C were stained with FM4-64 and samples were taken at the indicated time points. Control panel belongs to the same strain stained with the dye for 1 h in YPD medium. Bars, 1 μm.

**Table 1 jof-06-00355-t001:** *C. albicans* strain used in this work.

Strain	Genotype	Name in the Text	Reference
CAF2	*URA3/ura3*Δ*::imm434*	CAF2 (wt)	[30]
HI3-21	CAI4 *hog1::hisG/hog1::hisG-URA3-hisG*	*hog1*	[28]
CK43B-16	CAI4 *cek1::hisG/cek1::hisG-URA3-hisG*	*cek1*	[14]
E5	CAI4 *cek1::hisG-URA3-hisG/cek1::hisG* *hog1::hisG/hog1::hisG* *ARD1/ard1::FRT* pSAP2-FLP-URA3	*cek1 hog1*	[27]

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
