# Peer review of "Hog1 Controls Lipids Homeostasis Upon Osmotic Stress in Candida albicans"

_jof, 2020, doi:10.3390/jof6040355_

Round 1
Reviewer 1 Report
This MS discusses the role of Hog1 in the response to osmotic stress. It is concluded that this MAPK is involved in lipid homeostasis. After studying a hog1 mutant, it was noticed that this cell stores lipid droplets when under osmotic stress. This activity rises the cell permeability and delays the endocytic trafficking routes. Hence, it is here suggested that this contributes to the sensitivity to osmotic stress.
This work is interesting and seems to be well performed. However, several details are lacking and need to be indicated before acceptance. Among them several references are missing in some sentences and paragraphs and details in M&M (e.g. brands, country of manufacturers).
Introduction:
- Page 1, Line 29: “kinase” should begin with capital letter (as Mitogen Activated Protein);
- Page 1, Lines 29-38: “Signal transduction pathways mediated by Mitogen Activated Protein kinases…and induces the appropriate response.” – Reference(s) for this paragraph?
- Page 2, line 47: remove capital on “Quorum”;
Material and Methods:
- YPD medium: was prepared from the yeast, peptone and glucose or bought? In both cases, please indicate brands and countries of manufacture companies;
- Please indicate the approximate cells’ concentration for OD 0.1 and 1;
- Indicate all brands and country of the manufacturers of all materials and reagents used (e.g. NaCl);
- PBS: define abbreviation before using it. Molarity? Composition?
- Page 3, line 82: “100 to 300 colonies per plate in each strain.” Why this particular CFUs? Reference(as)?
- “Staining with propidium iodide” – reference(s) for the method? Also, this part needs to be correctly formatted;
- GraphPad: manufacturer and country?
Results and Discussion:
- Page 4, line 145: “Cell permeability …or due to an alteration of lipids in the membrane of the cells.” – reference(s) for this part?
- Line 161-163: “red and yellow (merge is observed in green) indicating that these phospholipid-rich intracellular membrane can be charged with neutral lipids such as triacylglycerols or cholesteryl esters...” - how was this concluded? Reference(s)?
- Figure 2B is distorted. Correct this;
- Figure 2 C – why aren’t the pair rows (fluorescence) coloured?
Discussion
- “Enzymes involved in fatty acid mobilization localized in internal lipid storage and peroxisomes.” – “are” is missing in the sentence?
- “In response to salt challenge, cerevisiae cells switch from a fermentative metabolism to fatty acid oxidation and mitochondrial respiration to cover the energetic needs during the environmental change. This process depends, in part, on the Hog1 activation. ” Is this from reference 44?
Author Response
Introduction:
- Page 1, Line 29: “kinase” should begin with capital letter (as Mitogen Activated Protein);
- The mistake has been corrected
- Page 1, Lines 29-38: “Signal transduction pathways mediated by Mitogen Activated Protein kinases…and induces the appropriate response.” – Reference(s) for this paragraph?
- Three references have been included at the end of the sentence.
- Page 2, line 47: remove capital on “Quorum”
- The capital has been removed;
Material and Methods:
- YPD medium: was prepared from the yeast, peptone and glucose or bought? In both cases, please indicate brands and countries of manufacture companies;
- The brands and countries of manufacture companies have been included in material and methods: Yeast strains were routinely grown in YPD medium (1% yeast extract (Conda/Pronadisa, Madrid, Spain), 2% bacteriological peptone (Conda/Pronadisa, Madrid, Spain), 2% glucose )(D(+)-glucose anhidra, PanReac AppliChem, Barcelona, Spain))
- Please indicate the approximate cells’ concentration for OD 0.1 and 1;
- The following information has been added to the text: An O.D. of 1 corresponds approximately to 2.5 x 107 cell/mL.
- Indicate all brands and country of the manufacturers of all materials and reagents used (e.g. NaCl)
- The following information has been included: For this purpose, NaCl (PanReac AppliChem, Barcelona, Spain)
- PBS: define abbreviation before using it. Molarity? Composition?
- The PBS (Phosphate-Buffered Saline) used for the experiments was PBS BD FACS Flow form Becton-Dickinson. This information has been included in material and methods section.
- Page 3, line 82: “100 to 300 colonies per plate in each strain.” Why this particular CFUs? Reference(as)?
- The CFU count allows estimating the number of live cultivable cells. Standard methods indicate to count plates containing 30 to 300 colonies. When we performed previous assays, we standardized the time and dilutions required to allow CFU quantification. Then, we select those conditions in which we got a representative number of colonies that in our case were between 100 and 300 colonies as indicated in the material and methods. The reference of the standard method , doi:DOI: 10.2105/SMWW.2882.188. has been included
- “Staining with propidium iodide” – reference(s) for the method? Also, this part needs to be correctly formatted;
- The reference of the methods is form De la Fuente et al, Yeast 1992, this and a more current reference ( Kwolek-Mirek and Zadrag-Tecza FEMS Yeast Res 2014) have been included and the paragraph has been re-formatted
- GraphPad: manufacturer and country?
- Following the suggestion of the reviewer the GraphPad Prim software has been correctly referenced: GraphPad Prism version 7.00 for Windows, GraphPad Software, San Diego, California USA, www.graphpad.com
Results and Discussion:
- Page 4, line 145: “Cell permeability …or due to an alteration of lipids in the membrane of the cells.” – reference(s) for this part?
- The following references have been included : DOI:10.1046/j.1198-743x.2001.00307.x and DOI 10.1099/jmm.0.010538-0, These papers describe that cell treated with antifungal that alter plasma membrane become rapidly PI positive. In our case, hog1 mutant cells grown under osmotic stress became PI+ before we detected no increase in CFUs count (we did not detect CFUs decrease) then, this PI permeabilization could be due alteration in the plasma membrane, subsequently in the lipids that form this plama membrane.
- Line 161-163: “red and yellow (merge is observed in green) indicating that these phospholipid-rich intracellular membrane can be charged with neutral lipids such as triacylglycerols or cholesteryl esters...” - how was this concluded? Reference(s)?
- We thank the critical revision of the manuscript since we realized that we have made a mistake. The sentence has been rephrased as follows: emitted fluorescence detected in green indicating that these phospholipid-rich intracellular membrane can be charged with neutral lipids such as triacylglycerols or cholesteryl esters in a higher amount compared to wild type cells [41]. The following reference has been included: https://doi.org/10.1371/journal.pone.0008499.
- We have also changed the figure legend : Red and green fluorescence is shown ( co-localization is seen in yellow)
- Figure 2B is distorted.
- Microscopy images have been changed to include scale bars. Confocal picture has been updated too.
- Figure 2 C – why aren’t the pair rows (fluorescence) coloured?
- The fluorescence pictures are not coloured because the camera that we used to capture the pictures provide black and white images, the colour can be given artificially but we decided to keep the original pictures.
Discussion
- “Enzymes involved in fatty acid mobilization localized in internal lipid storage and peroxisomes.” – “are” is missing in the sentence?
- The sentence has been corrected
- “In response to salt challenge, cerevisiae cells switch from a fermentative metabolism to fatty acid oxidation and mitochondrial respiration to cover the energetic needs during the environmental change. This process depends, in part, on the Hog1 activation. ” Is this from reference 44?
The reference has been included
Reviewer 2 Report
Herrero-de-Dios and collaborators present their work on the role of Hog1 in the control of lipid homeostasis under osmotic stress conditions in Candida albicans. Authors in a simple way show the effect of Hog1 and the MAPkinase Cek1 on the lipid composition and trafficking under high sodium stress. The manuscript is well written and the results are clear. Conclusions fit with the results obtained.
I could only suggest to the authors to include images showing that fluorescent dye FM4-64 is migrating to vacuoles and what is visualized under stress is alterations in vacuolar system by staining with CMAC.
Authors should include scale bars in the insets of microscopy images.
Author Response
We thank the suggestion of the reviewer and we will consider this information for future researches.
We have include scale bars in the microscopy images.